# Decision-making with speculative opponent model-aided value function factorization

## Abstract

In many real-world scenarios, teams of agents must coordinate their actions while competing against opponents. Traditional multi-agent reinforcement learning (MARL) approaches often treat opponents as part of the environment, causing controlled agents to overlook the impact of their adversaries. Opponent modeling can enhance an agent's decision-making by constructing predictive models of other agents. However, existing approaches typically rely on centralized learning with access to opponent data, and the process of extracting decentralized policies becomes impractical with larger teams. To address this issue, we propose the **D**istributional **S**peculative **O**pponent-aided **MIX**ing framework (DSOMIX), a novel value-based speculative opponent modeling algorithm that relies solely on local information—namely the agent's own observations, actions, and rewards. DSOMIX uses speculative beliefs to predict the behaviors of unseen opponents, enabling agents to make decisions based on local observations. Additionally, it incorporates distributional value decomposition models to capture a more granular representation of the agent's return distribution, improving the training process for the speculative opponent models. We formally derive a value-based theorem that underpins the training process. Extensive experiments across four challenging MARL benchmarks from MPE and Pommerman, demonstrate that DSOMIX outperforms state-of-the-art methods with superior performance and convergence.

## 1 Introduction

Multi-agent reinforcement learning (MARL) holds considerable promise to help address a variety of cooperative multi-agent problems, including cooperation, competition, and a mix of both (Hernandez-Leal et al., 2019; Zhang et al., 2021). In these dynamic environments, decision-making becomes increasingly complex, particularly when agents must account for interactions with other agents, both cooperative and adversarial. Among these, opponents—adversarial agents that hinder the controlled agents' objectives—play a crucial role, as their behaviors directly affect the transition dynamics experienced by the controlled agents. Many existing works normally regard the opponents (if exist) as a part of the environment and only use the controlled agents' information during training, such as MADDPG (Lowe et al., 2017), COMA (Foerster et al., 2018b), QMIX (Rashid et al., 2018) and soft Q learning (Wei et al., 2018), which results in the policy tending to be sub-optimal.

Some researchers argue that controlled agents should explicitly model the unknown goals and behaviors of opponents, rather than treating them as part of the environment, to enhance decision-making. This has led to the development of opponent modeling (Albrecht & Stone, 2018), where agents construct models to predict the behavior of other agents. However, many of these methods assume free access to opponents' information during training, including their observations and actions, which serve as ground truth (He et al., 2016; Foerster et al., 2018a; Raileanu et al., 2018; Tian et al., 2019; Papoudakis et al., 2021). However, obtaining the actual observations and actions of opponents may be impractical or expensive in many scenarios, such as poker (Papoudakis et al., 2020) and hidden and seek (Kamal et al., 2023). Even with complete knowledge of opponents' configurations, collecting detailed data on their observations, actions, and rewards becomes increasingly costly as the number of agents and task complexity grows. Consequently, controlled agents often need to rely solely on their own local information—such as their observations, actions, and rewards—to model opponents. Recent works have employed encoder-decoder architectures to extract representations

from this local information (Papoudakis et al., 2020; 2021), but they still rely on opponents' true data as the ground truth during training.

To address this issue, we propose a value-based adversary modeling framework for opponents, called the *Distributional Speculative Opponent-aided MIXing framework (DSOMIX)*. This framework predicts opponents' actions using local observations while maximizing the agent's utility. Specifically, at each time step, these speculative opponent models—so named because they do not rely on direct training signals from actual opponent behaviors—use the agent's observations as input to estimate the opponents' action distributions. The agent then samples multiple possible joint actions for the opponents, each of which is considered in the agent's decision-making process. The final action-value function is derived by incorporating these estimated opponent behaviors, allowing the agent to take the best possible response. To enhance the reliability of these speculative models, the quality of the agent's action-value function can be used as feedback during training. To further increase feedback, we employ a distributional decomposition module (Bellemare et al., 2017; Dabney et al., 2018b) to model the return distribution of the agent's action-value function. This approach provides richer information than a simple expected return, improving the agent's decision-making process.

In summary, the key contributions of this work is outlined as follows: (i) We introduce a speculative opponent model-aided value function factorization framework that leverages local information to infer unknown opponent behaviors. (ii) To the best of our knowledge, we are the first to utilize distributional value function factorization to guide the training of opponent models, providing a novel approach that could inspire future research. Additionally, we present a formal derivation, grounded in the value decomposition theorem, that supports the joint training of both the agent and speculative opponent models, strengthening the theoretical foundation of our approach. (iii) We empirically demonstrate that the proposed DSOMIX outperforms state-of-the-art methods on four challenging multi-agent tasks from the MPE (Lowe et al., 2017) and Pommerman (Resnick et al., 2018) environments. Our extensive experiments confirm that DSOMIX successfully learns reliable opponent models without access to opponents' true information, achieving superior task performance and faster convergence compared to baseline methods.

## 2 PRELIMINARIES

### 2.1 PARTIALLY OBSERVABLE MARKOV GAMES.

A *partially observable Markov game* (POMG) Lowe et al. (2017) of $n$ agents is formulated as a tuple $\mathcal{M} = \langle \mathcal{S}, \mathbb{O}, \mathcal{O}, \mathcal{A}, \mathcal{T}, \mathcal{R}, \gamma \rangle$. $\mathcal{S}$ is a set of states describing the possible configuration of all agents and the external environment. Also, each agent $i$ has its own observation space $\mathbb{O}_i \in \mathbb{O}$. Due to the *partial observability*, in every state $s \in \mathcal{S}$, each agent $i$ gets a correlated observation $o_i$ based on its observation function $\mathcal{O}_i : \mathcal{S} \to \mathbb{O}_i$ where $\mathcal{O}_i \in \mathcal{O}$. The agent $i$ selects an action $a_i \in A_i$ from its own action space $A_i \in \mathcal{A}$ at each time step, giving rise to a joint action $[a_1, \cdots, a_n] \in A_1 \times A_2 \times \cdots \times A_n$. The joint action then produces the next state by following the state transition function $\mathcal{T} : \mathcal{S} \times A_i \times \cdots \times A_n \to \mathcal{S}$. $\mathcal{R} = \{r_i\}$ is the set of reward functions. After each transition, agent $i$ receives a new observation and obtains a scalar reward as a function of the state and its action $r_i : \mathcal{S} \times A_i \to \mathbb{R}$. The initial state $s \in \mathcal{S}$ is determined by some prior distribution $p : \mathcal{S} \to [0, 1]$. Each agent $i$ aims to maximize its own total expected return $R_i = \mathbb{E}_{r^t \sim r_i(s_t, a_i^t), (s_t, a_i^t) \sim \tau} \sum_{t=0}^{T} \gamma^t r^t$, where $\gamma$ is the discount factor, $r^t$ is its sampled reward at time step $t$, $\tau$ is the trajectory distribution induced by the joint policy of all agents, and $T$ is the time horizon. Without loss of generality, we assume that the $n$ agents can be divided into $|M| \leq n$ teams, and each team has $q$ agents with $1 \leq q \leq n$. We consider the other teams as opponent agents which are controlled by a set of fixed policies. Note that a single agent can also form a team. In this paper, we assume agents from the same team fully cooperate and thus share the same reward function.

### 2.2 VALUE-BASED METHODS

The value-based method only consists of a *critic* to represent the action-value function with a deep neural network (Mnih et al., 2015). The $Q$ function can be recursively rewritten as $Q^\pi(s, a) = E_{s'}[r(s, a) + \gamma E_{a' \sim \pi}[Q_\theta^\pi(s', a')]]$. The *critic* learns parameters $\theta$ by sampling batches from the

replay memory and minimizing the TD error:

$$\mathcal{L}(\theta) = E_{s,a,r,s'}[(Q(s,a;\theta) - y)^2]$$

where $y = r + \gamma \max_{a'} Q(s', a'; \phi')$. $\theta'$ are the parameters of a target network that are periodically copied from $\theta$ and kept constant for a number of iterations. Although it can be directly applied to multi-agent settings by having each agent learn an independently optimal function (Tan, 1993), this approach does not adequately address the non-stationary engendered by the changing policies of other learning agents. In contrast, value decomposition methods aim to learn a joint value function, and the agents are trained in a centralized fashion and executed in a decentralized manner. To ensure consistency, the joint value function $Q_{jt}$ needs to satisfy the Individual-GlobalMax (IGM) principle (Rashid et al., 2018):

$$\arg\max_a Q_{jt}(\mathbf{o}, \mathbf{a}) = \begin{pmatrix} \arg\max_{a_1} Q_1(o_1, a_1) \\ \vdots \\ \arg\max_{a_n} Q_n(o_n, a_n) \end{pmatrix}, \tag{1}$$

where $\mathbf{o}$ is a joint action observation and $\mathbf{a}$ is a joint action. This work considers a most widely used value decomposition framework, called QMIX (Rashid et al., 2018), which incorporates a parameterized mixing network to compute the joint Q-value predicated on each agent's individual state-action value function: $Q_{jt}(\mathbf{o}, \mathbf{a}) = M(Q_1(o_1, a_1), \cdots, Q_n(o_n, a_n))$, where $M$ is a monotonic function that satisfies $\frac{\partial M}{\partial Q_n} \leq 0$. QMIX is trained with the objective of minimizing the DQN loss.

### 2.3 DISTRIBUTIONAL REINFORCEMENT LEARNING.

Distributional reinforcement learning (Bellemare et al., 2017) explicitly models the random return $Z^\pi(s, a)$ instead of its expectation. The distributional Bellman equation can be defined as,

$$Z(s, a) \overset{D}{=} r(s, a) + \gamma P^\pi Z(s, a), \tag{2}$$

where $\overset{D}{=}$ means the two sides of the equation are distributed according to the same law, $P^\pi Z(s, a) \overset{D}{=} Z(s', a')$ and $s' \sim P(\cdot|s, a), a' \sim \pi(\cdot|S')$. The distributional Bellman optimality operator is defined as: $\mathcal{T}^* Z(s, a) :\overset{D}{=} r(s, a) + \gamma Z(s', \arg\max_{a'} E[Z(s', a')])$.

Given some initial distribution $Z_0$, $Z$ converges to the return distribution $Z^\pi$ under $\pi$, contracting in terms of p-Wasserstein distance for all $p \in [1, \infty)$ under $\pi$; while $Z$ alternates between the set of optimal return distributions $\mathcal{Z}^* := \{Z^{\pi^*} : \pi^* \in \Pi^*\}$, where $\Pi^*$ denotes the set of optimal policies. The p-Wasserstein distance between the probability distributions of random variables $X, Y$ can be calculated by $W_p(X, Y) = (\int_0^1 |F_X^{-1}(\omega) - F_Y^{-1}(\omega)|^p d\omega)^{1/p}$, where $(F_X^{-1}, F_Y^{-1})$ are quantile functions of $(X, Y)$. The relationship between the cumulative distribution function (CDF) $F_X$ and the quantile function $F_X^{-1}$ (the generalized inverse CDF) of $X$ is formulated as $F_X^{-1}(\omega) = \inf\{x \in \mathbb{R} : \omega \leq F_X(x)\}, \forall \omega \in [0, 1]$, where $\omega$ represents the quantile. The expectation of $X$ expressed in terms of $F_X^{-1}(\omega)$ is defined as $E[X] = \int_0^1 F_x^{-1}(\omega) d\omega$. In (Dabney et al., 2018b), the value function can be modeled as quantile function $F^{-1}(o, a|\omega)$. Then, a pair-wise sampled TD error $\delta$ for two quantile samples $\omega, \omega' \sim U([0, 1])$ can be defined as,

$$\delta_t^{\omega, \omega'} = r + \gamma F^{-1}(o', a'|\omega') - F^{-1}(o, a|\omega). \tag{3}$$

The pair-wise loss $\rho_\omega^\kappa$ is then defined based on the Huber quantile regression loss $\mathcal{L}_\kappa$ (Dabney et al., 2018b) with threshold $\kappa = 1$, and can be formulated as,

$$\rho_\omega^\kappa(\delta^{\omega, \omega'}) = |\omega - \mathbb{I}\{\delta^{\omega, \omega'} < 0\}| \frac{\mathcal{L}_\kappa(\delta^{\omega, \omega'})}{\kappa}, \tag{4}$$

where the Huber loss $\mathcal{L}_\kappa(\delta^{\omega, \omega'}) = \frac{1}{2}(\delta^{\omega, \omega'})^2$ when $|\delta^{\omega, \omega'}| \leq \kappa$, otherwise, $\mathcal{L}_\kappa(\delta^{\omega, \omega'}) = \kappa(|\delta^{\omega, \omega'}| - \frac{1}{2}\kappa)$. Given $K$ quantile sample $[\omega_i]_{i=1}^K$ to be optimized with regard to $K'$ target quantile samples $[\omega_j]_{j=1}^{K'}$, the loss $\mathcal{L}(o, a, r, o')$ is defined as the sum of the pair-wise losses:

$$\mathcal{L}(o, a, r, o') = \frac{1}{K'} \sum_{i=1}^K \sum_{j=1}^{K'} \rho_{\omega_i}^\kappa(\delta^{\omega, \omega'_j}). \tag{5}$$

## 3 METHODOLOGY

In this section, we introduce the proposed Distributional Speculative Opponent-aided MIXing method (DSOMIX). We first describe the overall framework of DSOMIX in section 3.1. Then we provide details and the theoretical foundation of the proposed method in section 3.2. Finally, a practical implementation of DSOMIX is presented in Section 3.3.

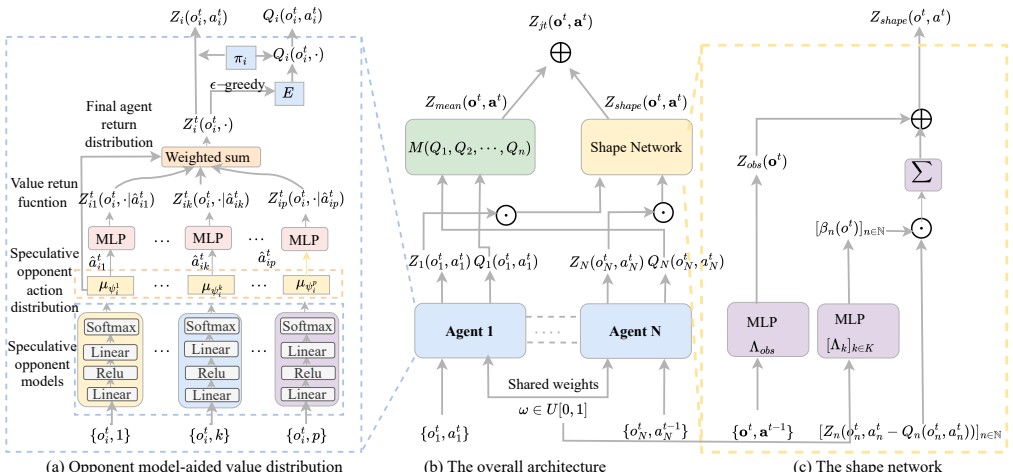

(a) Opponent model-aided value distribution    (b) The overall architecture    (c) The shape network

Figure 1: An illustration of our DSOMIX network architecture. (a) Speculative opponent models: Each model contains $p$ speculative opponent models, which take the local observation $o_i$ and opponent index $k$ as inputs to predict the opponents' actions. Then, the action value function network takes the joint predicted action $\{\hat{a}_i^t\}$ together with the $o_t^i$ and $a_i^{t-1}$ as input, and outputs a distribution over the agent $i$'s own actions, which is weighed according to the probabilities of predicted opponents' actions. (b) The DSOMIX framework consists of a mixing network $M(Q_1, Q_2, \cdots, Q_N)$ and a shape network $\Phi$ for decomposing the deterministic part $Z_{mean}$ (i.e., $Q_{jt}$) and the stochastic part $Z_{shape}$ of the total return distribution $Z_{jt}$. (c) The shape network contains parameter networks $\Lambda_{obs}(\mathbf{o}, \mathbf{a}^{t-1})$ and $[\Lambda_n(\mathbf{o})]_{n \in \mathbb{N}}$ for generating $Z_{obs}(\mathbf{o}^t)$ and $[\beta_n(\mathbf{o}^t)]_{n \in \mathbf{N}}$.

### 3.1 THE OVERALL FRAMEWORK

Our training follows the CTDE setting, which grants access to the observations and actions of the team under our control. Note that, however, we **do not know** the observation and actions of the opposing team during training. Figure 1 depicts the overall framework of our DSOMIX algorithm. For each agent, there is one agent network that represents its individual value function $Z_i(o_i^t, a_i^t)$. We first allocate a collection of speculative opponent models for each agent. For $p$ opponents, agent $i$ employs $p$ individualized conjectural opponent models, wherein each model is instantiated as a distinct neural network. These models process the agent's localized observational data $o_i^t$ and sampled actions at each timestamp to ascertain the presumptive action distributions of the opponents. Utilizing these distributions, we proceed to sample multiple opponent joint actions $\hat{a}_i^t$, each of which is fed into the controlled agent. The controlled agent's final value function, $Z(o_i^t, \cdot)$, is the weighted sum of the agent's outputs where each weight is the sampling probability of the corresponding opponent's joint action.

By this design, the decision-making process of the controlled agent is strongly intertwined with its speculative opponent models. Specifically, the speculative opponent model aims to infer potential joint actions, representing the agent's conjectures concerning the opponents' prospective moves. The agent $i$'s own value function is subsequently determined as a composite function of the outputs from the speculative opponent model and the observations, effectively assimilating the agent's tactical considerations with the inferences drawn from the opponent models by performing a weighted aggregation. To enhance predictive precision, the agent samples multiple opponents' joint

actions. Intriguingly, the opponent models' training regimen is predicated on the rewards obtained by the agent's action alone, eschewing reliance on the opponents' genuine actions and underscoring outcome-oriented feedback loops. Meanwhile, the agent value function role is to evaluate the return distribution associated with the agent's action, preferring a distributional perspective over singular expected value estimations.

## 3.2 Speculative Opponent Model-aided Value Function Factorization

Suppose agent $i$ has $p$ opponents in the game, which are represented as $p$ independent speculative opponent models. As shown in Figure 1(a), there is one agent network that receives the current individual observation $o_i^t$ and the last action $a_i^{t-1}$ as inputs and outputs its individual value function. Let $\mu_{\psi_{ik}}$ be parameterized by trainable parameters $\psi_{ik}$ and $\psi_i = \{\psi_{ik}\}$ be the set of parameters of all speculative opponent models maintained by agent $i$. Each speculative opponent model $\mu_{\psi_{ik}}$ takes as input the observations $o_i^t$ and the index of the opponent and outputs a probability distribution over the opponent's action space. From this distribution, the agent samples each opponent $k$'s action to construct the joint predicted action $\hat{a}_i^t = [\hat{a}_{i1}^t, \ldots, \hat{a}_{ip}^t]$. After the agent in the controlled team sample their actions, each controlled agent $i$'s value function network $Z_{\phi_i, \psi_i}$ takes the joint predicted action together with the observation $o_i^t$ as input, and computes a return distribution $Z_{\phi_i}(o_i^t, \cdot | \{\hat{a}_{ikl}\})$. We conduct the sampling process multiple times and aggregate the resulting return distribution into the final individual return distribution $Z_{\phi, \psi}(o_i^t, \cdot)$. Therefore, the return distribution with speculative opponent models of agent $i$ can be defined as:

$$Z_{\phi_i, \psi_i}(o_i^t, \cdot) = \sum_l Z_{\phi_i}(o_i^t, \cdot | \{\hat{a}_{ikl}\}) \prod_{k=1}^{p} \mu_{\psi_{ik}}(\hat{a}_{ikl} | o_i^t). \tag{6}$$

Since IGM is necessary for value function factorization, a distributional factorization that satisfies IGM is required for factorizing stochastic value functions with speculative opponent models. We then provide a theorem to show that the distributional factorization with speculative opponent model is sufficient to guarantee the IGM condition. We noticed that a finite number of individual stochastic utilities $[Z_{\phi_i, \psi_i}(o_i, a_i)]_{i \in \mathbb{N}}$ satisfies the distributional version of IGM for a stochastic joint action value function $Z_{jt}(\mathbf{o}, \mathbf{a})$ under $\mathbf{o}$, since $[\mathbb{E}[Z_{\phi_i, \psi_i}(o_i, a_i)]]_{i \in \mathbb{N}}$ satisfies IGM for $\mathbb{E}[Z_{jt}(\mathbf{o}, \mathbf{a})]$, that is:

$$\arg\max_a \mathbb{E}[Z_{jt}(\mathbf{o}, \mathbf{a})] = \begin{pmatrix} \arg\max_{a_1} E[Z_{\phi_1, \psi_1}(o_1, a_1)] \\ \vdots \\ \arg\max_{a_N} E[Z_{\phi_N, \psi_N}(o_N, a_N)] \end{pmatrix}. \tag{7}$$

Next, we present the formal derivation of the speculative opponent model-aided value decomposition theorem.

**Theorem 3.1.** *Consider a deterministic joint action-value function $Q_{jt}$, determined by a factorization function $M$, a stochastic joint action-value function $Z_{jt}$:*

$$Q_{jt}(\mathbf{o}, \mathbf{a}) = M(Q_1(o_1, a_1), \cdots, Q_N(o_N, a_N)),$$

*such that $[Q_n]_{n \in \mathbb{N}}$ satisfy IGM for $Q_{jt}$ under $\mathbf{o}$. The following distributional factorization:*

$$\begin{aligned} Z_{jt}(\mathbf{o}, \mathbf{a}) &= \mathbb{E}[Z_{jt}(\mathbf{o}, \mathbf{a})] + (Z_{jt}(\mathbf{o}, \mathbf{a}) - \mathbb{E}[Z_{jt}(\mathbf{o}, \mathbf{a})]) \\ &= Z_{mean}(\mathbf{o}, \mathbf{a}) + Z_{shape}(\mathbf{o}, \mathbf{a}) \\ &= Q_{jt} + \Phi(Z_1(o_1, a_1), \cdots, Z_N(o_N, a_N)) \end{aligned} \tag{8}$$

*is sufficient to guarantee that $[Z_n]_{n \in \mathbb{N}}$ satisfy IGM for $Z_{jt}$ under $\mathbf{o}$, where $\mathbb{E}[\Phi] = 0$.*

This theorem reveals that the choice of factorization function $M$ determines whether IGM holds, regardless of the choice of $\Phi$, as long as $\mathbb{E}[\Phi] = 0$. Under this setting, any differentiable factorization function of deterministic variables can be extended to a factorization stochastic value function with speculative opponent model. We provide the proof of Theorem B.1 in the appendix.

In this work, we employ an implicit quantile network (IQN) (Dabney et al., 2018a), which is an effective way to learn an implicit representation of the return distribution, to approximate $Z_i$. IQN is a deterministic parametric function trained to parameterize samples from a quantile distribution $\omega \sim U([0, 1])$ to the respective quantile values of a target distribution. In execution, the action with the

largest expected return $Q_i(o_i^t, a_i^t)$ is chosen, where $Q_i(o_i^t, a_i^t)$ can be approximated by calculating the mean of the sampled return through $N'$ quantile samples $\omega_j \sim U([0,1])$, $\forall j \in [1, N']$. The expression is as follows:

$$Q_i(o_i^t, a_i^t) \approx \frac{1}{N'} \sum_{j=1}^{N'} Z_i^j(o_i^t, a_i^t).$$

Therefore, we can obtain the agent return distribution $Z_i(o_i^t, a_i^t)$ and respective agent value function $Q_i(o_i^t, a_i^t)$. Then we construct the joint action return distribution by considering each agent's individual return distribution. To satisfy monotonicity, the joint action return distribution $Z_{jt}(\mathbf{o}^t, \mathbf{a}^t)$ should be decomposed into its deterministic part $Z_{mean}$ and stochastic part $Z_{shape}$ based on the Mean-Shape decomposition (Sun et al., 2021). As shown in Figure 1(b), we approximate $Z_{mean}$ by a factorization network $M$, and a shape network $\Phi$ was used to approximate $Z_{shape}$. The factorization function must accurately decompose the expectation of $Z_{jt}$ to adhere to monotonicity constraints. On the other hand, the shape function is allowed to roughly factorize the shape of the $Z_{jt}$. Therefore, $Z_{jt}$ can be approximated by,

$$Z_{jt}(\mathbf{o}^t, \mathbf{a}^t) = M(Q_1, \ldots, Q_n) + \sum_{i=1}^{n} (Z_i - Q_i) = M(Q_1, \cdots, Q_n) + \Phi(Z_1, \cdots, Z_N). \quad (9)$$

The factorization network can be trained end-to-end to minimize the DQN loss, which is defined as,

$$\mathcal{L}_{DQN}(\theta) = \sum_{b=1}^{B} [(\hat{Z}_{mean}^b - Z_{mean}(\mathbf{o}, \mathbf{a}; \theta))^2], \quad (10)$$

where $B$ is the batch size of transitions sampled from the replay buffer, and $\theta = (\phi, \psi)$ are the parameters of the agent network. $\hat{Z}_{mean} = r + \gamma max_{\mathbf{a}'} Z_{mean}(\mathbf{o}', \mathbf{a}'; \theta')$ and $\theta'$ are the parameters of the target network. Then, the shape network $\Phi$ can be implemented by a large IQN composed of multiple IQNs, optimized through the Quantile Huber loss, as defined in equation 5.

### 3.3 A Practical Implementation of DSOMIX

In this section, we introduce the practical implementation of the proposed method. According to the above analysis, computing the marginal distribution $\rho_{\theta_i, \psi_i}$ and final action return distribution $Z_{\phi, \psi}$ can be exponentially costly concerning the dimensionality of the opponents' action space. Formally, each agent $i$'s opponent $k$ has $|A_{ik}|$ actions. Then calculating $Z_{\phi, \psi}$ requires traversing $|A_{i1}| \times \cdots \times |A_{ip}|$ combinations of opponents' predicted actions, which quickly becomes intractable as $p$ increases. Therefore, we apply a sampling trick that samples a set of actions $\hat{a}_{ik} = (\hat{a}_{ik1}, \cdots, \hat{a}_{ikl})$ from the output of the speculative opponent model $\mu_{\psi_{ik}}$ for each opponent $k$, where $l$ controls the size of sampled actions. Thus the final agent return distribution can be approximated by:

$$\bar{Z}_{\phi_i, \psi_i}(o_i, a_i) = \sum_l Z_{\phi_i}(a_i | o_i, \{\hat{a}_{ikl}\}) \prod_{k=1}^{p} \mu_{\psi_{ik}}(\hat{a}_{ikl} | o_i).$$

For the deterministic part $Z_{mean}$, we introduce the mixing network, which is a feed-forward neural network taking the agent network outputs as input and mixes them monotonically, producing the values of $Z_{mean}$. For the stochastic part $Z_{shape}$, we use the property of quantile mixture to approximate the shape function $\Phi$. The quantile function $F_{shape}^{-1}$ of $Z_{shape}$ can be approximated by:

$$F_{shape}^{-1}(\mathbf{o}, \mathbf{a} | \omega) = F_{obs}^{-1}(\mathbf{o} | \omega) + \sum_{n \in \mathbb{N}} \beta_n((F_n^{-1}(o_n, a_n | \omega) - Q_n(o_n, a_n))), \quad (11)$$

where $F_{obs}^{-1}(\mathbf{o} | \omega)$ and $[\beta_n(\mathbf{o})]_{n \in \mathbb{N}}$ are respectively generated by function approximators $\Lambda_{obs}(\mathbf{o} | \omega)$ and $[\Lambda_n(\mathbf{o})]_{n \in \mathbb{N}}$, satisfying constraints $\beta_n(\mathbf{o}) \leq 0$, $\forall n \in \mathbb{N}$ and $\int_0^1 F_{obs}^{-1}(\mathbf{o} | \omega) d\omega = 0$. The term $F_{obs}^{-1}$ models the shape of an additional observation-dependent utility.

In practice, the algorithm initially generates training data utilizing the current DSOMIX. Subsequently, the objective of DSOMIX is computed from the generated data. Finally, the optimizer

updates the parameters of the agent network accordingly. The updated agent network are then employed in the next training iteration. We parallelize the training, which is a common technique to reduce the training time (Iqbal & Sha, 2019). In such cases, the training data is gathered from all parallel environments, and actions are sampled and executed in respective environments concurrently. We summarize the training procedure in Algorithm 1.

## 4 EXPERIMENT

To comprehensively evaluate the proposed algorithm, we conduct experiments on two widely adopted partially observable multi-agent environments with various settings and tasks, namely the Predator-prey environment and the Pommerman environment, and compare with extensive baselines. From the experiments, we aim to answer the following questions: (1): Does the DSOMIX yield superior performance than SOTA baseline methods (Figure 2(a), 2(b), 2(d), and 2(e) and Table 1)? (2): Are the main components of DSOMIX, i.e., the SOM and distribution value function, necessary and effective (Figure 2(c), Figure 2(e), and Figure 3)? (3): What is the connections between speculative opponent model (SOM) and distributional value function (Figure 4)?

### 4.1 THE ENVIRONMENT SETUP

**Setup for the predator-prey environment.** In predator-prey environment (Figure 5(a)), the player controls multiple collaborating predators aiming to catch swifter preys within 500 iterations. Each prey possesses a health value of 10. A predator moving within a given range of the prey lowers the prey's health by 1 point per time step. Lowering the prey's health to 0 can kill the prey. If at least one prey remains after 500 iterations, the prey team wins. All agents can select from five distinct movement actions. At the start of the game, $L$ gray landmarks are randomly placed in the environment as obstacles to potentially impede the agents' paths. Each predator receives the relative positions and velocities of the agents, along with the landmarks' positions as an observation. We evaluate our method in two scenarios. The first scenario includes three predators and one prey, which we denote as `PP-3v1`. Another scenario includes five predators and two preys, denoted as `PP-5v2`. The number of landmarks is 2 in both settings.

**Setup for the Pommerman environment.** This environment involves four agents, and each agent can either move in one of four directions, place a bomb, or do nothing. As shown in Figure 5(b), a state is represented as an image consisting of three different square grids, i.e., empty, wooden, or rigid. An empty grid permits any agent to enter it, while a wooden grid is inaccessible but destructible by a bomb. In contrast, a rigid grid is unbreakable and impassable. When a bomb is placed in a grid, it will explode after 10 time steps. The explosion will destroy any adjacent wooden grids and kill any agents within 4-grids away from the bomb. If all agents belonging to one team die, the team loses the game. The game will be terminated after 1000 steps no matter whether there is a winner team or not. Agents get a $+1$ reward if their team wins and $-1$ reward otherwise. The experiments are carried out in two different scenarios. The first scenario consists of four agents fight against each other, which we denote as `Pomm-FFA`. The other scenario is a team match with two teams of two agents, which we denote as `Pomm-Team`. The details of all the environments and scenarios are provided in Appendix C.1.

### 4.2 BASELINES AND ALGORITHM CONFIGURATION.

**Baselines.** We compare DSOMIX with the most well-known value decomposition algorithms, QMIX (Rashid et al., 2018). QMIX is aligned with our setting where opponents' true information is not available. In addition to the original QMIX algorithm, we further introduce two more baselines based on QMIX by integrating QMIX with our speculative opponent models and distributional variant, respectively. In specific, let OMIX denote the baseline that combines QMIX with the speculative opponent model, while DMIX is the variant that combine QMIX with distributional value decomposition. OMIX and DMIX allow us to evaluate the impact of speculative opponent models and distributional critic, respectively. To comprehensively verify the performance of DSOMIX, we also compare it with MAPPO (Yu et al., 2022), which has achieved state-of-the-art (SOTA) performance in many environments. To evaluate the accuracy of learned speculative opponent model, we

include an "upper-bound" (UB) baseline, which substitutes the actual opponent policy as the ground truth of opponent models during training DSOMIX.

**Algorithm configuration.** For `PP-3v1` and `PP-5v2`, we train the networks for $E = 35600$ episodes. Each agent adopts $\epsilon$-greedy action selection strategy, with $\epsilon$ linearly from 1.0 to 0.05 over 100 episodes, keeping it constant for the rest of the learning. We set $\gamma = 0.95$ for all experiments. All neural networks are trained using `Adam` optimizer (Kingma & Ba, 2015) with a learning rate of $2.5e - 4$. The replay buffer consists of the latest 100 episodes, from which we uniformly sample a batch of size 32 for training. The target network is updated every 100 episodes. We have conducted a study on the impact of sample size $l$. We observe that when $l$ is small, increasing it improves the performance obviously. However, a large sample size only brings marginal benefits while requiring too much computation. Therefore, we set the sample size of `PP-5v2` as $l = 10$. The others follow the default setting of Pytorch (Paszke et al., 2017).

The configuration for `Pomm-FFA` and `Pomm-Team` is generally the same as that of the predator-prey games. The learning rate for agent network is both $2.5e-5$. We parallel 16 environments during training and the number of forward step is $T_f = 5$, that is, we update the networks after collecting 5 steps of data from 16 environments at each iteration. The total number of training episodes is $E = 624,000$. We set the sample size $l$ in `Pomm-FFA` and `Pomm-Team` as 80 and 25, respectively. The details of the algorithm configurations are provided in the Appendix C.2. All experiments are carried out in a machine with Intel Core i9-10940X CPU and a single Nvidia GeForce 2080Ti GPU. We will make all our data and codes public after the work is accepted.

### 4.3 EXPERIMENT RESULTS

#### 4.3.1 ANALYSIS OF THE AVERAGE RETURN.

We first compare the overall performance of DSOMIX with baselines in the four games, i.e., `PP-3v1`, `PP-5v2`, `Pomm-FFA`, and `Pomm-Team`. The results are summarized in Figure 2(a), 2(b), 2(d), and 2(e). We measure the performance in terms of the approximate expected return, and each model is trained with 8 random seeds. Specifically, for `PP-3v1` and `PP-5v2`, we evaluate all the methods with 100 test episodes after every 100 iteration of training, and report the mean (solid lines) and the standard deviation (shaded areas) of the average returns over eight seeds. Similarly, for `Pomm-FFA` and `Pomm-Team`, all methods are evaluated with 200 test episodes after every $1,000$ training iteration. The experiment results demonstrate that our method not only obtains a higher average return but also achieves a faster convergence speed than baseline methods. The comparative analysis between DSOMIX and the baselines demonstrates the effectiveness of our speculative opponent models in learning opponent models with only local information. The experimental results reveal that the returns of DSOMIX are comparable to the upper bound (UB) and consistently outperform the four other baselines, with a faster convergence speed and lower variance. These findings confirm that our method is an effective variant for addressing the challenges posed by POSG without access to opponents' information. It is important to note that QMIX treats opponents as a part of the environment, while DSOMIX explicitly models the opponents and integrates them into its agent decision-making process. Although the speculative opponent models are not reliable initially, the reward signals train them to provide reliable information. As a result of providing supplementary information for decision-making, DSOMIX demonstrates superior performance compared to QMIX. In addition, the performance of OMIX indicates that incorporating the speculative opponent models can boost the performance of the QMIX. DMIX outperforming QMIX demonstrates that turning the expected return estimation into a distributional value function also helps to make decisions. These two results show that both the speculative opponent models and the distributional value function have benefits for improving performance alone. Furthermore, when compared to MAPPO, a state-of-the-art policy gradient algorithm, DSOMIX outshines in both sample efficiency and overall performance. This suggests that learning distinct return distributions and accurately anticipating the actions of unknown opponents can indeed significantly elevate performance.

#### 4.3.2 ABLATION STUDY

Throughout these ablation studies, we use game `PP-5v2` as a demonstration. We first verify that the speculative opponent models (SOM) truly help to make better decisions, and we perform a pair of experiments. The first one employs trained and fixed opponent models, named

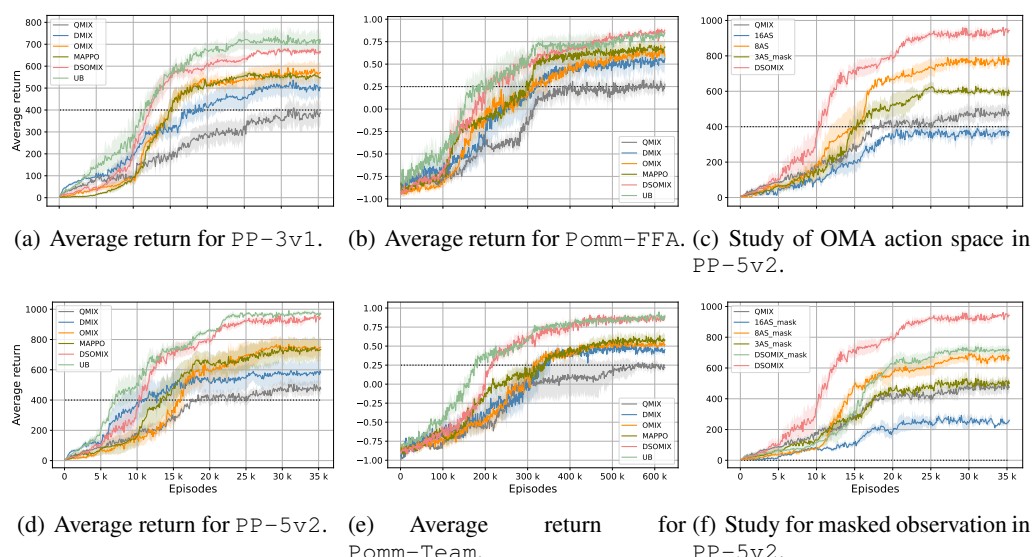

(a) Average return for `PP-3v1`. (b) Average return for `Pomm-FFA`. (c) Study of OMA action space in `PP-5v2`.

(d) Average return for `PP-5v2`. (e) Average return for `Pomm-Team`. (f) Study for masked observation in `PP-5v2`.

Figure 2: Results of performance evaluation and ablation studies in DSOMIX.

"Trained SOM", while the second one uses randomly initialized, and fixed opponent models instead, named "Non-trained SOM". The average returns are plotted in Figure 3. It is obvious that the DSOMIX with trained SOM learns faster than the version without trained models. It can be indicated that the agent can infer the behaviors of its opponents and take advantage of this prior knowledge to make better decisions, especially at the beginning stage of the learning procedure. It shows that SOM can provide reliable information for better decision-making.

While the previous results show that SOM improves the decision-making quality, one may wonder whether the improvement really results from the opponent action prediction output by the SOM. To investigate whether arbitrary trainable SOM can introduce improvement, we conduct an ablation study for the SOM to see how it affects the training of DSOMIX, and the results over 8 random seeds are shown in Figure 2(c). We change the output dimension $d$ of speculative opponent models to 3, 8, and 16 respectively while retaining other configurations. Note that the opponent action space size is 5. There-

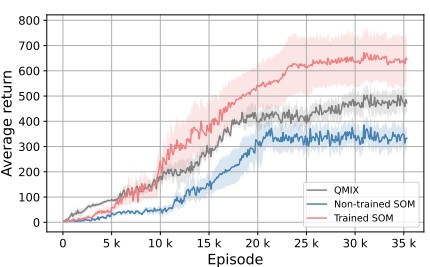

Figure 3: Study for trained SOM.

fore, the speculative opponent models in the three new settings output some conditional information instead of opponent action predictions. We denote these setting as "3AS", "8AS", and "16AS" respectively in Figure 2(c). It can be observed that the original DSOMIX exhibits superior learning performance in terms of both speed and effectiveness when compared to the other variants.We also notice that the performance of $d = 3$ and $d = 8$ are consistently better than that of $d = 16$ for the entire training period. It implies that the speculative opponent model can infer more reliable information when its output dimension is close to the opponent action space size.

Given the above results, a question arises in our mind is why the SOM perform the best when their architectures are designed to output opponent action prediction? One potential reason is that the opponents sometimes appear in the observations of the controlled agents and thus, the local observations can occasionally convey the state-changing information of the opponents. In this case, the SOM that output opponent action prediction can indeed learn better from the received observations. To verify this hypothesis, we conduct another experiment where we mask out the opponent information when the agent observes the opponents and retains the other configurations. The results are denoted as "X_mask" where "X" means the original algorithm setting. Figure 2(f) shows that when masking out the opponent information, the performance of all algorithms declines, which

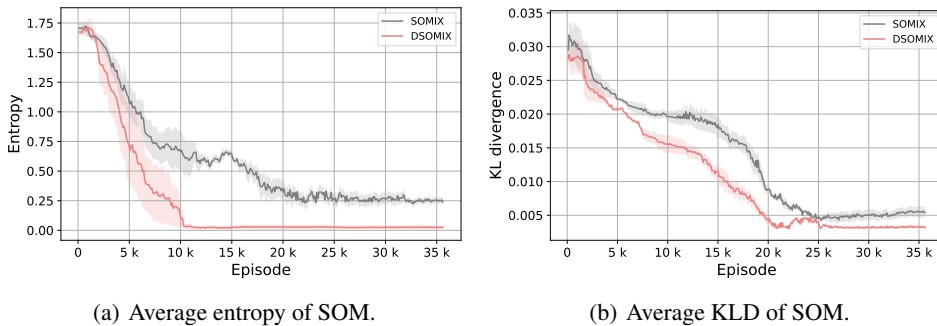

(a) Average entropy of SOM.  (b) Average KLD of SOM.

Figure 4: Impacts of the distributional value function for game `PP-5v2`.

means that the opponent model learning indeed benefits from the information contained in the local observations. It shows that when masking out the opponent information, the performance of "DSOMIX_mask" is weak to "DSOMIX", which means that DSOMIX indeed learns something from the observations that contain the opponent information (i.e., the observed coordinate values in our experiments). However, it still outperforms QMIX and "QMIX_mask", which shows that the SOM helps to make better decisions in our method.

### 4.3.3 Exposing connections between SOM and distributional value function.

To prove that the speculative opponent models are more accurate in predicting opponent actions with a distributional value function, Two metrics were utilized to evaluate the improvement: : 1). We calculate the average entropy of the predicted probability distribution over opponent actions, which serves as an indicator of prediction confidence. Lower average entropy implies greater certainty on the part of agents regarding the predicted opponent's actions. 2). We compute the Kullback-Leibler divergence (KLD) (Kullback & Leibler, 1951) between the predicted and the true probability distribution over opponent actions. The KLD is the direct measure of the distance between the speculative opponent models and the true opponent policies. A smaller KLD value signifies a higher degree of similarity between the opponent models and the true policies. Note that we only use the true opponent policies for evaluation. We do not use them to train DSOMIX. From Figures 4(a) and 4(b), we conclude that the distributional value function can increase the training speed (faster descent) and improve the reliability and confidence (lower KLD and entropy) of the opponent models. The second argument, i.e., the distributional value function helps the agent to identify actions with more rewards, is supported by the results in Figure 2, where the performance of DMIX is better than QMIX. Note that the expected returns of DMIX are still lower than DSOMIX, which implies that the integration of SOM and distributional value function is essential for our method. The ablation studies, along with the OMIX and DMIX baselines, demonstrate that both speculative opponent models and distributional value function play crucial roles in our algorithm. The distributional value function contributes to the development of higher-quality speculative opponent models, which, in turn, enhance the overall performance by facilitating better speculative opponent models.

## 5 Conclusion

This work proposes a distributional speculative opponent-aided mixing framework (DSOMIX), a novel value-based speculative opponent modeling algorithm that relies solely on local information. In our methods, the speculative opponent models receive as input the controlled agents' local observations which predict the opponents' unknown behaviors when opponents' information is unavailable. With the guidance of the distributional value function, we manage to train the agent network and speculative opponent models effectively. Extensive experiments demonstrate that our methods not only obtain a higher average return but also achieve a faster convergence speed. The ablation studies and the baselines prove that the SOM and distributional value function are both essential parts for our algorithm. That is, the distributional value function leads to a higher-quality SOM and in turn, the better SOM helps to improve the overall performance. However, our work assumes opponents have fixed strategies in the environment. Further research on how such models could be used for non-stationary opponents would be of interest. For future work, we will study how to

model opponents with dynamic strategies only using the local information, which is more practical and challenging in the real-world settings.

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

## A  RELATED WORK

### A.1  MULTI-AGENT REINFORCEMENT LEARNING

In recent years, numerous MARL algorithms have been developed, leveraging reinforcement learning techniques to jointly train agents within multi-agent systems (Hernandez-Leal et al., 2019). Early research focused on independent learning (IL), where each agent learned in isolation, treating other agents as part of the environment (Tan, 1993; Mnih et al., 2016; Schulman et al., 2017). However, this approach struggled with the non-stationarity introduced by interacting learning agents. In contrast, centralized training with decentralized execution (CTDE) allows for information sharing during training while maintaining policies conditioned only on local observations during execution (Lowe et al., 2017). Within CTDE, centralized policy gradient methods employ decentralized actors and a centralized critic optimized through shared agent information (Lowe et al., 2017; Foerster et al., 2018b; Yu et al., 2022). Another category, value decomposition methods, focuses on decomposing joint state-action value functions into individual value functions (Sunehag et al., 2017; Rashid et al., 2018). However, these methods often treat opponents as part of the environment, neglecting their explicit influence and leading to sub-optimal learning outcomes.

### A.2  LEARNING OPPONENT MODEL

To improve decision-making, researchers argue that controlled agents must infer the objectives and actions of opponents, leading to the development of opponent modeling. Recent advances in deep learning architectures have greatly accelerated progress in this field (Albrecht & Stone, 2018; He et al., 2016; Hong et al., 2018; Raileanu et al., 2018; Albrecht & Stone, 2017; Rabinowitz et al., 2018; Yang et al., 2019; Zheng et al., 2018; Wen et al., 2019; Zintgraf et al., 2021; Tian et al., 2019; Liu et al., 2019). However, most of these approaches assume access to opponents' observations and actions during both training and execution. More recent studies (Papoudakis et al., 2020; 2021) suggest that such access is often infeasible, particularly in large-scale applications. These works propose learning opponent models based solely on the agent's local information, eliminating the need for opponent data during execution. However, these approaches still require opponents' true information during training. The challenge of modeling opponent behaviors when such information is unavailable during training remains an open problem. To address this, we propose a speculative opponent model-aided value function factorization framework that infers unknown opponent behavior using local information during both training and execution.

### A.3  DISTRIBUTIONAL REINFORCEMENT LEARNING

Distributional reinforcement learning (RL) models the return distributions rather than expected returns, using these distributions to optimize policies (Bellemare et al., 2017; Dabney et al., 2018a;b). Numerous studies have demonstrated that distributional RL outperforms classical RL methods (Barth-Maron et al., 2018; Tessler et al., 2019; Singh et al., 2020; Yue et al., 2020). Inspired by its success in single-agent scenarios, recent works (Lyu & Amato, 2020; Hu et al., 2020; Sun et al., 2021) have extended distributional RL to multi-agent settings. Notably, DFAC (Sun et al., 2021) bridges distributional RL and value function factorization, introducing the Mean-Shape Decomposition and quantile mixture in value decomposition. This approach mitigates instability arising from the exploration of learning agents during training. In this work, we propose a speculative opponent model-based framework that integrates distributional value function factorization with speculative opponent modeling. Our framework employs distributional RL to evaluate the quality of the agent's value function, guiding the training of speculative opponent models. Moreover, our findings demonstrate the potential of distributional RL to inspire further developments in this field.

## B  SPECULATIVE OPPONENT MODEL-AIDED VALUE FUNCTION FACTORIZATION THEOREM

In this part, we present the details of the speculative opponent model-aided value function factorization theorem. We present proofs of our theorem introduced in the main text as follows.

**Theorem B.1.** *Consider a deterministic joint action-value function $Q_{jt}$, determined by a factorization function $M$, a stochastic joint action-value function $Z_{jt}$:*

$$Q_{jt}(\mathbf{o}, \mathbf{a}) = M(Q_1(o_1, a_1), \cdots, Q_N(o_N, a_N)),$$

*such that $[Q_n]_{n \in \mathbb{N}}$ satisfy IGM for $Q_{jt}$ under $\mathbf{o}$. The following distributional factorization:*

$$
\begin{aligned}
Z_{jt}(\mathbf{o}, \mathbf{a}) &= \mathbb{E}[Z_{jt}(\mathbf{o}, \mathbf{a})] + (Z_{jt}(\mathbf{o}, \mathbf{a}) - \mathbb{E}[Z_{jt}(\mathbf{o}, \mathbf{a})]) \\
&= Z_{mean}(\mathbf{o}, \mathbf{a}) + Z_{shape}(\mathbf{o}, \mathbf{a}) \\
&= Q_{jt} + \Phi(Z_1(o_1, a_1), \cdots, Z_N(o_N, a_N))
\end{aligned}
\tag{12}
$$

*is sufficient to guarantee that $[Z_n]_{n \in \mathbb{N}}$ satisfy IGM for $Z_{jt}$ under $\mathbf{o}$, where $\mathbb{E}[\Phi] = 0$.*

*Proof.* For any given random variable $Z_{jt}$, there exist a unique decomposition defined as

$$Z_{jt} = \mathbb{E} + (Z_{jt} - \mathbb{E}[Z]) = Z_{mean} + Z_{shape}, \tag{13}$$

where $Var(Z_{mean}) = 0$ and $\mathbb{E}(Z_{shape}) = 0$. Therefore, $Z_{jt}(\mathbf{o}, \mathbf{a})$ can be written as:

$$
\begin{aligned}
&\arg\max_{\mathbf{a}} \{\mathbb{E}[Z_{jt}(\mathbf{o}, \mathbf{a})]\} \\
&= \arg\max_{\mathbf{a}} \{\mathbb{E}[Z_{mean}(\mathbf{o}, \mathbf{a}) + Z_{shape}(\mathbf{o}, \mathbf{a})]\} \\
&= \arg\max_{\mathbf{a}} \{\mathbb{E}[Z_{mean}(\mathbf{o}, \mathbf{a})] + \mathbb{E}[Z_{shape}(\mathbf{o}, \mathbf{a})]\} \\
&= \arg\max_{\mathbf{a}} \{\mathbb{E}[M(Q_1(o_1, a_1), Q_N(o_N, a_N))] + \mathbb{E}[\Psi(Z_1(o_1, a_1), \cdots, Z_N(o_N, a_N))]\} \\
&= \arg\max_{\mathbf{a}} \{M(Q_1(o_1, a_1), Q_N(o_N, a_N)) + 0\} \\
&= \arg\max_{\mathbf{a}} \{M(Q_1(o_1, a_1), Q_N(o_N, a_N))\} \\
&= \begin{pmatrix} \arg\max_{a_1} Q_1(o_1, a_1) \\ \vdots \\ \arg\max_{a_n} Q_n(o_n, a_n) \end{pmatrix}
\end{aligned}
\tag{14}
$$

Therefore, we can obtain that

$$
\arg\max_{a} \mathbb{E}[Z_{jt}(\mathbf{o}, \mathbf{a})] = \begin{pmatrix} \arg\max_{a_1} E[Z_{\phi_1, \psi_1}(o_1, a_1)] \\ \vdots \\ \arg\max_{a_N} E[Z_{\phi_N, \psi_N}(o_N, a_N)] \end{pmatrix}.
\tag{15}
$$

The above derivation demonstrates that $Z_{\phi_i, \psi_i}(o_i, a_i)_{i \in \mathbb{N}}$ satisfy IGM for $Z_{jt}$ under observation $\mathbf{o}$.

## C EXPERIMENT

### C.1 ENVIRONMENTAL SETTINGS

**Predator-prey:** The states, observations, actions, state transition function, and reward function of each agent is formulated below by following the POMG convention.

- **The states and observations.** A grid world of size $x \times x$, e.g. Figure 5(a) is a state of size $7 \times 7$ containing four predators and two preys. The observation of agent $i$ is the coordinates of its location, its ID, and the coordinates of the prey $k$ relative to $i$ in $l \times l$ view, if observed.
- **Actions space.** Any agent, either predators or preys, has five actions, i.e. [up, down, left, right, no-op] where the first four actions means the agent moves towards the corresponding direction by one step, and no-op indicates doing-nothing. All agents move within the map and can not exceed the boundary.
- **State transition $\mathcal{T}$.** The new state after the transition is the map with updated positions of all agents due to agents moving in the grid world. The termination condition for this task is when all preys are dead or for 100 steps.

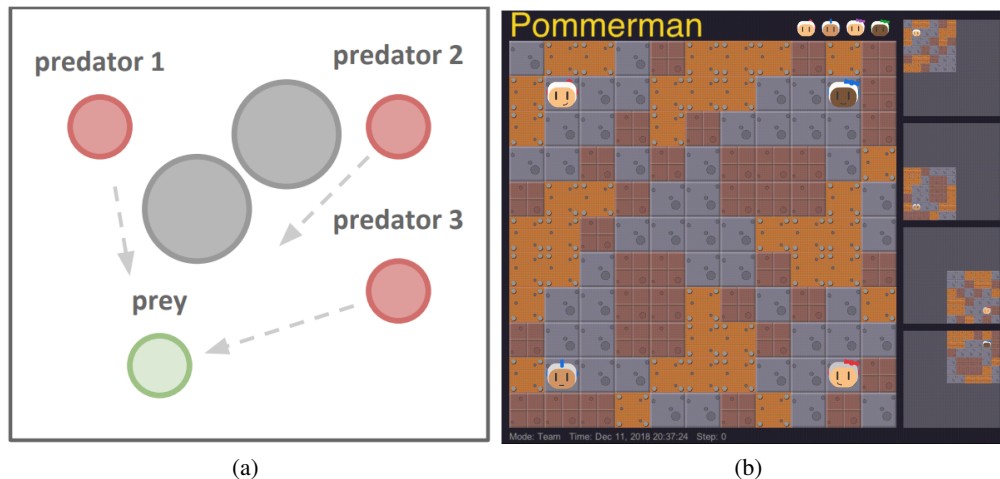

(a)                                                          (b)

Figure 5: **State visualization of benchmark environments.** (a) The state of a `PP-5v2` game is represented as a grid world, where blue vertices and red vertices denote predators and preys respectively. (b) The image-based state for the Pommerman environment.

- **Rewards** $\mathcal{R}$**.** All agents move within the map and can not exceed the boundary. Since the predators cooperate with each other, they share the team reward. The predators share a reward of 5 if two or more of them catch the prey simultaneously, while they are given a negative reward of -0.5 if only one predator catches the prey.

**Pommerman:**

- **The states and observations.** At each time step, agents get local observations within their field of view $5 \times 5$, which contains information (board, position,ammo) about the map. The agent obtain the information of the Blast Strength, whether the agent can kick or not, the ID of their teammate and enemies, as well as the agent's current blast strength and bomb life.

- **Actions space.** Any agent chooses from one of six actions, i.e. $[\text{up}, \text{left}, \text{right}, \text{down}, \text{stop}, \text{bomb}]$. Each of the first four actions means moving towards the corresponding directions while `stop` means that this action is a pass, and `bomb` means laying a bomb.

- **Rewards** $\mathcal{R}$**.** In `Pomm-Team`, the game ends when both players on the same team have been destroyed. It ends when at most one agent remains alive in `Pomm-FFA`. The winning team is the one who has remaining members. Ties can happen when the game does not end before the max steps or if the last agents are destroyed on the same turn. Agents in the same team share a reward of 1 if the team wins the game, they are given a reward of -1 if their team loses the game or the game is a tie (no teams win). They only get 0 reward when the game is not finished.

## C.2 ALGORITHM CONFIGURATION

For `PP-3v1` and `PP-5v2`, the speculative opponent network is multi-layer perceptrons (MLP) with 3 hidden layers of dimensionality 64. In DSOMIX, the agent network of each controlled agent $i$ is a DRQN with a recurrent layer comprised of a GRU with a 64-dimensional hidden state, with a fully-connected layer before and after. The factorization network is a feed-forward neural network that takes the agent network outputs of $Q_i$ as input and mixes them monotonically, producing the values of $Z_{mean}$. The weights of the factorization network are produced by separate hypernetworks. Each hypernetwork consists of a single linear layer, followed by an absolute activation function, to ensure that the mixing network weights are non-negative. The shape network is implemented by a large IQN composed of multiple IQNs. We optimize the IQNs with $N = 32$ quantile samples. The final bias $\beta$ is produced by a 2-layer hypernetwork with a ReLU non-linearity.

---

**Algorithm 1** General Training Procedure of DSOMIX

---

**Require:** A POMG environment *env*, a back-propagation optimizer `Opt`, number of episodes $E$.
**Require:** Initialize parameters of $\theta$ of the agent network, speculative opponent model, factorization network, and shape network;
**Require:** Initialize parameters of $\theta$ of the target network of agent, speculative opponent model, factorization network, and shape network;
**Require:**
**Training phase:**
1: **for** $e = 1, \ldots, E$ **do**
2:     $t \leftarrow 1$; Reset *env* to obtain initial observations $\mathbf{o}^1$.
3:     **while** *env* is not done **do** // `Generate data`
4:         Sample actions $\mathbf{a}^t$, where $a_i^t \in \mathbf{a}^t$ follows $a_i^t \sim Z_{\theta_i, \psi_i}(a_i^t | o_i^t)$.
5:         Execute $\mathbf{a}^t$ to obtain joint rewards $\mathbf{r}^t$ and new observations $\mathbf{o}^{t+1}$.
6:         Store transition data $(\mathbf{o}^t, \mathbf{a}^t, \mathbf{r}^t, \mathbf{o}^{t+1})$.
7:         **for** steps in training steps **do** // `Update network`
8:             Sample a min-batch $\mathcal{D}'$ from replay buffer $\mathcal{D}$.
9:             Calculate the distributional value function $\{Z_i(o_i, a_i)\}_{i \in \mathbb{N}}$ with collected data.
10:             Updates the agent network by minimizing the TD loss (10) and QR loss (5).
11:             Update $\bar{\theta}$: $\bar{\theta} \leftarrow \theta$;
12:         **end for**
13:     **end while**
14: **end for**

---

The configuration for `Pomm-FFA` and `Pomm-Team` is generally the same as that of the predator-prey games. However, here the agent value function network is a convolutional neural network (CNN) with 4 hidden layers, each of which has 64 filters of size $3 \times 3$, as the observations are image-based. Between any two consecutive CNN layers, there is a two-layer MLP of dimension 128. The learning rate for $Z_{\phi_i}$ is both $2.5e - 5$.

## C.3 ANALYSIS OF THE LEARNING SPEED

To demonstrate the efficiency of our approach, we compare the relative learning speed of our methods and baselines with that of QMIX (without loss of generality). This evaluation is defined by the formula $LS = E_{P_{QMIX}}/T$, where $P_{QMIX}$ represents the optimal performance for QMIX (indicated by the black dashed line in Figures 2(a), 2(b), 2(d), 2(e)) (in the main paper), and $E_{P_{QMIX}}$ denotes the episode count at which different methods reach this benchmark performance (including QMIX, DMIX, OMIX, MAPPO, DSOMIX, and UB). To give an example, in the scenario `PP5v2`, with a total of $T = 356000$ training episodes, DSOMIX achieves the same performance as $P_{QMIX}$ at episode 10750, resulting in a relative learning speed of 30.2%. As summarized in Table 1, the learning speed of DSOMIX consistently surpasses all other methods by a large margin over all the tested scenarios. Furthermore, it is worth noting that DSOMIX exhibits a convergence speed comparable to UB (the baseline trained with ground-truth opponents' information). The exceptional data efficiency of DSOMIX can be attributed to the agent networks in effectively guiding the opponent modelling process. Meanwhile, the speculative opponent model, in turn, aids the agent's policy in making more informed decisions.

Table 1: The learning speed of different methods.

|        | QMIX  | DMIX  | OMIX  | MAPPO | DSOMIX    | UB        |
|--------|-------|-------|-------|-------|-----------|-----------|
| `PP3v1` | 70.3% | 50.1% | 49.2% | 35.3% | **30.5%** | **29.9%** |
| `PP5v2` | 73.1% | 48.2% | 42.8% | 39.5% | **30.2%** | **25.3%** |
| `FFA`   | 71.2% | 50.1% | 49.4% | 41.8% | **28.1%** | **26.1%** |
| `Team`  | 74.1% | 42.7% | 46.5% | 37.9% | **33.6%** | **27.4%** |

