# OpenReview forum: "Decision-making with speculative opponent model-aided value function factorization"
_ICLR.cc/2025/Conference — Submitted to ICLR 2025_

### Official Review · Reviewer_s7Qt · 2024-10-28

**Soundness:** 2
**Presentation:** 2
**Contribution:** 1
**Rating:** 3
**Confidence:** 4

**Summary:**

This paper introduces DSOMIX (Distributional Speculative Opponent-aided MIXing), a novel framework for multi-agent reinforcement learning (MARL) that addresses the challenge of opponent-aware decision-making using decentralized policies. By leveraging speculative opponent models, DSOMIX predicts adversarial behaviors based on local observations without requiring access to opponent data during training. Additionally, it incorporates distributional value function factorization to provide more granular return estimates, improving decision quality and convergence speed. The paper is well-supported by theoretical derivations and extensive experiments across benchmarks like MPE and Pommerman, demonstrating superior performance over baselines. While effective, the approach assumes fixed opponent strategies, leaving room for future work on dynamic, non-stationary opponents.

**Strengths:**

DSOMIX framework does not rely on access to opponent data during training.

**Weaknesses:**

1. Although the proposed approach is fundamentally an opponent modeling method, the paper does not compare its performance with any established opponent modeling baselines. This lack of comparison makes it difficult to assess the true effectiveness of DSOMIX relative to existing opponent modeling techniques.
2. The paper argues in the introduction that prior opponent modeling methods rely on opponent data, which prevents them from obtaining decentralized policies. However, this claim is inaccurate, as there are opponent modeling approaches that can achieve decentralized policies while still utilizing opponent data during training. This misrepresentation weakens the justification for the novelty of the proposed method.
3. This paper is poorly written, e.g., a pseudo algorithm may help.

**Questions:**

How can you support the claim "However, existing approaches typically rely on centralized learning with access to opponent data, and the process of extracting decentralized policies becomes impractical with larger teams." in the abstract?

---

### Official Review · Reviewer_F2Vp · 2024-11-01

**Soundness:** 3
**Presentation:** 3
**Contribution:** 2
**Rating:** 5
**Confidence:** 3

**Summary:**

This paper builds upon the foundation laid by [1] (though not explicitly cited) to propose DSOMIX, a speculative opponent modeling framework that enables MARL agents to make decisions based on local information alone.

**Strengths:**

This paper builds on previous work by incorporating value decomposition, enhancing the team coordination game-solving capabilities. Experimental results also demonstrate that the method outperforms Qmix.

**Weaknesses:**

The methodological contribution is relatively incremental, as it primary contribution is to combine value decomposition with prior work.

**Questions:**

1. The experimental comparisons seem limited. Given that DSOMIX includes opponent behavior modeling and requires multiple samples per state, it’s unsurprising that it outperforms Qmix when using episodes as the x-axis. Are there additional results using other metrics, such as wall clock time? Also, have comparisons been made with more recent methods?
2. Since this work builds on [1], is there an ablation study comparing against [1]?

**Typos:**

In the caption for Fig. 1, "An illustration of our **DOMAC** network architecture" mistakenly uses "DOMAC," which is the method name from [1], instead of this paper's "DSOMIX".

**Reference**:

[1] Sun, Jing, et al. "Decision-Making With Speculative Opponent Models." *IEEE Transactions on Neural Networks and Learning Systems* (2024).

---

### Official Review · Reviewer_UVCP · 2024-11-03

**Soundness:** 3
**Presentation:** 2
**Contribution:** 2
**Rating:** 5
**Confidence:** 3

**Summary:**

This paper introduces a method called distributional speculative opponent-aided mixing framework(DSOMIX), which is built upon QMIX, distributional Q, and has a speculative opponent modeling module. Their experiments in Pommerman and predator-prey show their methods achieve higher returns and faster convergence speed than the baseline methods. Their ablation study shows the distributional aspect of the value network and speculative opponent models are necessary for the framework.

**Strengths:**

The paper is a novel combination of DFAC(QMIX+distributional RL) and speculative opponent modeling, considering the conditional probability distribution on opponent’s potential actions. Their empirical experiment section studies effect of introduced modules (SOM).

**Weaknesses:**

(1) The distributional RL + QMIX aspect was introduced in the prior work [1], and Theorem 3.1 appears to be the same as Theorem 2 in [1] but this theorem is introduced in method part, potentially indicating it is an original contribution. Please exiplicitly discuss how Theorem 3.1 is different or related to the Theorem 2 in [1].

[1] Sun, W. F., Lee, C. K., & Lee, C. Y. (2021, July). DFAC framework: Factorizing the value function via quantile mixture for multi-agent distributional Q-learning. In *International Conference on Machine Learning* (pp. 9945-9954). PMLR.

**Questions:**

(1) What makes SOM+Distributional RL converges faster than OMIX and DMIX? Can you provide a more detailed analysis or ablation study specifically comparing the convergence rate of DSOMIX, OMIX and DMIX?  This could help isolate the factors contributing to faster convergence and provide deeper insights into the synergies between SOM and Distributional RL.

(2) How is the opponent model trained? For example, can you provide loss function used for training the opponent model, the specific training procedure, and how the opponent model interacts with the main agent during training. This would provide a more comprehensive understanding of the opponent modeling approach and its integration into the overall framework.

---

### Official Review · Reviewer_wcoy · 2024-11-04

**Soundness:** 2
**Presentation:** 2
**Contribution:** 2
**Rating:** 5
**Confidence:** 4

**Summary:**

This work proposes DSOMIX, Distributional Speculative Opponent-aided MIXing, a multi-agent reinforcement learning (MARL) approach that incorporates opponent modelling into the distributional value function factorization (DFAC) framework applied to the QMIX approach.

**Strengths:**

Using the additional information provided by distributional returns to also train opponent models is an interesting idea and novel contribution. Also, demonstrating that no additional knowledge apart from the local observation is enough for such auxiliary models is a valuable insight for this line of research.

**Weaknesses:**

I believe the contribution heavily relies and builds on the distributional value function factorization (DFAC) framework [1], both in terms of theory (the mean-shape decomposition, IGM and DIGM equivalence) and algorithmic design (IQN based implementation, the DMIX baseline), but this is not properly acknowledged. Please see questions Q1 and Q2, to address this issue.

Another concern regards the scalability of the approach, since each learning agent will consider _p_ opponent models. The ablation studies regarding the number of outputs for the opponent models is interesting, since knowing the opponents actions space size is not always possible. I have further questions related to this below, see Q3 - Q5.

Finally, a crucial limiting factor for the contribution of the work is the fact that the modelled opponents have fixed policies. See Q7.

[1] Sun, W. F., Lee, C. K., See, S., & Lee, C. Y. (2023). A unified framework for factorizing distributional value functions for multi-agent reinforcement learning. Journal of Machine Learning Research, 24(220), 1-32.

Minor remarks:
- I advise to keep the consistency of the colors for the approaches across figures (Fig 2, DSOMIX is either red or green)
- Fix notation inconsistencies, example critic parameters are denoted as $\phi$ (line 107) and then referred to as $\theta$ in lines 110 - 112
- typos, for example:
    - line 113 having each agent learns -> learn
    - line 130 Distributions RL explicitly model -> models
    - line 323 the objectives of DSOMIX is -> objective is or objectives are
    - line 357 An empty grid permits any agents -> agent
    - line 370 QMIX is align -> aligned
    - Figure 2 DOMAC? I assume it should be DSOMIX

**Questions:**

Q1. Can you please clarify how Theorem 3.1 differs from Theorem 2 in [1]? I do not see how the addition of the speculative opponent models in the individual agents affects the distributional value function factorisation theorem.

Q2. Can you clarify how or if the DMIX baseline used in this work differs from the DMIX in [1]?

Q3. Can you motivate the choice of training separate models for each opponents, inside each of the learning agents? Did you consider the option of training one single joint model of the opponents in each of the learning agents, that can capture a compressed representation of the opponents behaviour?

Q4. What are the computational costs, compared to DMIX?

Q5. Can you expand further on the sampling process for the opponents joint actions? That seems to be an important bottleneck of the approach, especially since it needs to be done multiple times.

Q6. In the algorithm configuration $\epsilon$ starts at 1. Is this value decayed?

Q7. Do you have any insights if the assumption of only observing information regarding the opponents through the local observations would be enough to still capture informative models in the case in which the opponents would also undergo a learning process?

---

### Meta-Review · Area_Chair_ETVX · 2024-12-22

**Metareview:**

This paper proposes incorporating opponent modeling into QMIX. The main concerns raised by the reviewers focus on the presentation and the lack of novelty:

- Insufficient Acknowledgment and Comparison: The work heavily builds upon or is inspired by several prior studies (as noted by Reviewers wcoy and s7Qt). However, the current presentation lacks adequate acknowledgment of or comparisons with those works.

- Questionable Novelty: Reviewer UVCP highlighted that Theorem 3.1, which is claimed as a novel contribution, appears to replicate results from previous studies. The authors did not respond to this point, further raising doubts about the originality of the theoretical contributions.

- Clarity Issues: Both Reviewer UVCP and Reviewer s7Qt found the paper poorly written and struggled to understand the proposed algorithm, indicating significant room for improvement in presentation quality.

**Additional Comments On Reviewer Discussion:**

This paper proposes incorporating opponent modeling into QMIX, but the reviewers raised concerns about its presentation and lack of novelty. First, the work heavily builds upon or is inspired by prior studies, yet it lacks sufficient acknowledgment and comparison with those works, as noted by Reviewers wcoy and s7Qt. Second, Reviewer UVCP questioned the novelty of Theorem 3.1, suggesting it replicates results from previous studies, and the authors' lack of response to this critique further undermines the originality of their theoretical contributions. Finally, both Reviewer UVCP and Reviewer s7Qt found the paper poorly written, with significant clarity issues that made it difficult to understand the proposed algorithm, highlighting the need for substantial improvements in presentation.

---

### Decision · Program_Chairs · 2025-01-22

Reject